# Causal Concept Identification in Open World Environments

**Moritz Willig**[†,1]     **Matej Zečević**[1]     **Jonas Seng**[1]     **Florian Peter Busch**[1,2]

[1] Computer Science Department, AIML, TU Darmstadt, Germany
[2] Hessian Center for AI (hessian.AI)
[†]correspondence: `moritz.willig@cs.tu-darmstadt.de`

## Abstract

The ability to continually discover novel concepts is a core task in open world learning. For classical learning tasks new samples might be identified via manual labeling. Since this is a labor intensive task, this paper proposes to utilize causal information for doing so. Image data provides us with the ability to directly observe the physical, real-world appearance of concepts. However, the information presented in images is usually of noisy and unstructured nature. In this position paper we propose to leverage causal information to both structure and causally connect visual representations. Specifically, we discuss the possibilities of using causal models as a knowledge source for identifying novel concepts in the visual domain.

**Overview.** Section (1) motivates continuous concept discovery using causal mechanisms. Section (2) outlines a path to continually advance the discovery of visual concepts using causal structures. Section (3) outlines practical issues encountered in (1) and (2) and discusses future steps.

## 1 Motivation

Modern machine learning systems need to process large amounts of annotated image data to identify visual concepts. While the resulting models achieve impressive results and push the limits of the field, they lack human curiosity and fall short in their ability to perform lifelong learning. One drawback of such approaches is the necessity to provide a supervision signal for the image data. While modern approaches utilize large amounts of data available from the internet, the trained models fall short on training data for niche domains and might adapt harmful biases from the data. Continual learning approaches are interested in *continuously discovering novel concepts* to help machine learning models improve and adapt to new environments. A key challenge in this regard is to provide sufficient amounts of accurately annotated data. For this purpose the field of causality can help to provide the required supervision by leveraging the structure of causal mechanisms.

**How do humans discover novel concepts?** Machine learning models are usually trained by presenting them with randomly sampled image-label pairs. The presented samples usually do not leverage connections to other already known concepts. Because of this, machine learning models are expected to identify visual concepts from ground up. This is in contrast to the way humans discover the world. When presented with novel concepts we usually start out with some initial knowledge. We relate novel concepts to pre-existing knowledge and therefore *continually* advance our understanding of the world (Chen and Liu 2018; Flesch et al. 2022).

**Supervision for open world discovery** Labeled data is scarce for many applications in machine learning. Some modalities, such as text data might be available in great abundance, while other modalities such as (annotated) image data are often lacking. When training on image data there are common types of supervision signals: Most commonly, images are assigned a label out of a set of fixed labels or categories (e.g. LeCun (1998); Deng et al. (2009)). Training on such data usually provides a clear supervision signal with regard to the task of interest. However, fixing the set of labels also restricts the set of possible concepts to provide feedback on. Including new concepts is a tedious process that possibly requires the collection of new data and (re)labeling of the data set. More recently, data sets of image-description pairs are used for training (Schuhmann et al. 2022). Collected images are no longer restricted to a certain set of labels, but are rather paired-up with an associated free-text description of the image contents. This helps our task of continuous concept discovery that requires a supervision signal that is able to describe newly appearing concepts. Additionally, using language as a supervision signal aligns better with human intuition and allows to express relations between objects/concepts in the image. As a trade-off way between fixed feature vectors and arbitrary text descriptions, we can make use of relational data. Applying supervision from manually annotated (Speer, Chin, and Havasi 2017) or learned (Willig et al. 2022; Long, Schuster, and Piché 2022) knowledge graphs combines the benefits of explicitly stating involved concepts while retaining information of relations between the individual concepts.

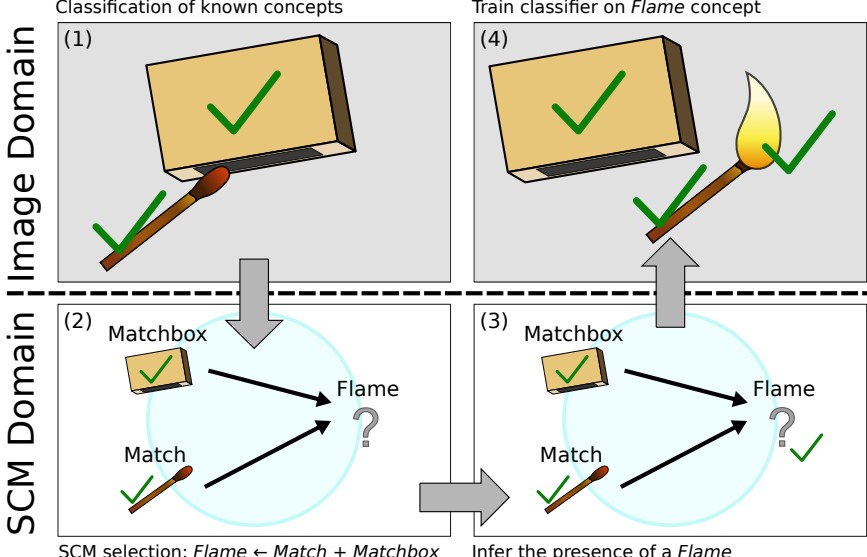

Figure 1: Visual concept discovery using structural causal models. (1) A classifier identifies the concepts *Match* and *Matchbox* in an image. (2) An SCM (blue circle) that covers the concepts *Match* and *Matchbox* is selected from a knowledge base. The SCM contains the additional concept of a *Flame* which is yet unknown to the classifier. (3) Evaluation of the structural equations indicate the presence of a flame. (4) The classifier is retrained to detect the previously unknown *Flame* concept.

**Why do we want causal concept discovery?** Causality is concerned with identifying processes that are underlying real world observations. As such, structural causal models (SCM) are designed to model the causal relationship between different concepts. Whenever we detect some concepts of a given SCM, we can utilize the graph structure to infer the presence of other, possibly unknown, concepts. Definitions of a 'concept' might vary, depending on the specific use case. In this paper we will use the term to capture the range from low-level features, such as color or texture, to complex composed entities, e.g. cities. In the following we outline a possible approach to utilize such causal knowledge for continual discovery (shown in Figure 1). Assume that a system is able to detect matches and matchboxes in images, but has not yet seen a flame. In Phase 1 the classifier will identify the known concepts match and matchbox, but is unable to identify a flame. However, we might have knowledge about the causal effect of sliding a match along the striking surface of the matchbox. We can express this knowledge in a causal graph without having to provide image samples (Phase 2). Now, whenever we observe the interaction of striking matches on a matchbox, we infer the presence of a flame (Phase 3) and in consequence train our classifier on the new concept (Phase 4). This approach helps us in two ways: (1) Since the causal model specifies the exact conditions under which a particular concept is to be expected, we can actively steer our discovery process towards those instances and discover concepts more efficiently. (2) Secondly, causality helps us with disentangling concepts and ruling out confounding factors, which may not be possible with arbitrary relational feedback. While some objects might only appear in strongly correlated settings we know the *true causal factors* from the SCM.

## 2 Causal Concept Discovery

Discovering concepts under causal supervision provides a way to incrementally discover and learn novel concepts in the image domain. For every observation, we search for an SCM that contains already known concepts and follow the causal graph to infer the presence of novel concepts. With each newly learned concept we are able to identify more concepts and in consequence use more causal graphs for inference. This continually broadens the scope of our models and discovers previously unknown concepts.

**Structural Causal Models.** As we use SCMs to *ground* and discover novel concepts in the visual domain, we follow the Pearlian notion of Causality (Pearl 2009). An SCM is defined as a 4-tuple $\mathcal{M} := \langle \mathbf{U}, \mathbf{V}, \mathcal{F}, p(\mathbf{U}) \rangle$ where the so-called structural equations $v_i \leftarrow f_i(\mathrm{pa}_i, u_i) \in \mathcal{F}$ assign values to the respective endogenous variables $V_i \in \mathbf{V}$ based on the values of their parents $\mathrm{Pa}_i \subseteq \mathbf{V} \setminus V_i$ and the values of their respective exogenous variables $\mathbf{U}_i \subseteq \mathbf{U}$. In particular, any SCM induces a *causal graph* which represents the causal structure from causes to effects.

**Bootstrapping.** SCMs are specifically tailored to represent information about causal systems. However, in practice we may not be able to explicitly provide a list of concepts $C$ learned by the model and might even encounter catastrophic forgetting of already learned concepts (French 1999). This poses a practical problem as it introduces additional uncertainty to our system. For our theoretical discussion, we assume for now that we can reliably detect all concepts that we have already discovered during our process.

For starting up our discovery process, we assume an initial set of known concepts $C_0$ to be given, which can be reliably detected. This intial knowledge might come from training

on manually annotated data sets or other pre-trained models (Lin et al. 2014; Krizhevsky, Sutskever, and Hinton 2012; Minderer et al. 2022). Additionally we assume to be given a source of SCMs that encode our causal knowledge about the world.

**Boundaries of Discovery.** Open world settings provide an infinite stream of concepts to discover (Chen and Liu 2018). Like for human curiosity, we are not interested in learning random concepts, but in utilizing our existing causal knowledge to efficiently discover concepts that stay close to our already existing knowledge. As such, we define the *causal frontier* as the set of SCMs that contain at least one concept of $C_i$. Importantly this gives us a method to a priori determine which concepts can and can not be discovered. Given the set of initial concepts $C_0$ and the set of given causal graphs, it follows that we can only discover those concepts for which we can find a chain of causal graphs, such that for any two adjacent SCMs we get a non-empty overlap between their set of variables.

**Discovering concepts.** At this point in our process we can start to advance our causal frontier by continually learning new concepts from observations. As a first step, we identify all known concepts that are present in a new observed image and select the causal graphs that contain those concepts. However, discovering that a concept is contained in the set of endogenous variables of a causal graph does not suffice to infer the presence of the causal system. Some common concepts, such as color, might appear as parameters in many causal graphs. Apples, for example, are typically colored green or red. As such, the concept of color *parameterizes* the observed object. However, it is not suited to infer the type of an object, as being colored red or green does not make an object an apple. Detecting the typical 'apple-like shape' however would be a strong indicator for the concept. Therefore, we are interested in SCMs for which we discover indicator variables or *actual causes* (Halpern 2016) that are necessary to infer the presence of a causal system.

Another important insight is the fact that some causal structures may only be identifiable with the help of interventions (Pearl 2009; Bareinboim et al. 2022). Consider, for example, a scenario of two independent variables, $A$ and $B$, whose appearance are determined by a third variable $C$. As a consequence, $A$ and $B$ are either present at the same time or not, and we have no way to disentangle them. Causality can help to detect such situations. In consequence we can actively intervene on the system and identify the individual concepts.

## Preconditions for Causal Concept Discovery

While supervision using relational (causal) structures trades off rigid label vectors and unstructured text supervision, we still need to overcome a number of obstacles for practical use.

As a special form of relational concept discovery, causal concept discovery relies on the assumption, that a causal process involving a given concept exists and that this process is identifiable in the data. This restricts the possible area of application, but acquires a stronger feedback in return. Coming back to our match striking example, we find a similar "Lighting Match" entry in ConceptNet[1] which describes the relational cause-effect relations between a match, matchbox and flame.

Given, not only the relational causal structure, but exact additional structural equations, will strengthen our discovery signal. Describing not only the existence of a flame but the process of starting out small and getting bigger over time gives us a directly measurable sequence. Even without proper calibration between the SCM 'flame size' variable to the actual observed flame size in the image, we observe two variables with equivariant behaviour over time, increasing the probability of a correct match between the SCM and the actual 'unknown' visual concept of a flame.

## 3    Challenges and Future Steps

In the previous sections we outlined the high-level idea of causally guided concept discovery. For this section we now continue to discuss the challenges that may arise in practice.

**Challenge 1: Identifying causal paths.** One problem with identifying unknown variables from SCMs is the fact that we may not know how causes and effects are interacting in the real world. In our initial example of lighting a match we might follow the physical process of the objects interacting. For other examples we might assume to only observe sparse changes. While we can continue to come up with more heuristics for specific problems, we have to recognize that the current formalizations of causality are not well suited to trace causal effects in their underlying systems.

**Challenge 2: Abstracting concepts.** Another challenge towards identifying unknown concepts from SCMs arises from the fact that causal systems are often modeled using high-level relations. Because of that we might encounter several low-level entities on the way from cause to effect the are not modeled in the SCM. In order to be able to identify these concepts, we need to consider abstractions and refinements of SCMs (Beckers and Halpern 2019; Rubenstein et al. 2017). This means, that we have to come up with ways of identifying intermediate concepts or refine the given SCMs to better reflect the abstraction level of our observations.

**Summary and Outlook:** In this position paper we highlighted the strengths and challenges of continuous causal concept discovery. We presented ways on how to leverage causal structures to guide concept discovery and identify novel concepts. While we primarily focused on the application of causal knowledge to discover open world concepts, the inverse problem of inferring causal knowledge from open world settings is also still to be discussed. Identifying relevant concepts and connecting them in a way that leads to meaningful causal concepts poses a challenging problem on its own that is yet to be solved. For future applications, we might consider combined approaches that discover visual concepts via causal guidance while simultaneously refining their causal knowledge using observations.

---

[1]https://conceptnet.io/c/en/lighting_match

**Acknowledgments**   The authors thank the anonymous reviewers of the Bridge program for their valuable feedback. Furthermore, the authors acknowledge the support of the German Science Foundation (DFG) project "Causality, Argumentation, and Machine Learning" (CAML2, KE 1686/3-2) of the SPP 1999 "Robust Argumentation Machines" (RATIO). This work was supported by the Federal Ministry of Education and Research (BMBF; project "PlexPlain", FKZ 01IS19081). It benefited from the Hessian research priority programme LOEWE within the project WhiteBox, the HMWK cluster project "The Third Wave of AI" (3AI) & the National High-Performance Computing project for Computational Engineering Sciences (NHR4CES).

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
