# OpenReview forum: "Causal Concept Identification in Open World Environments"
_AAAI.org/2023/Bridge/CCBridge — AAAI23 Bridge Continual Causality_

### Official Review · Reviewer_na2S · 2022-11-24
**Brief comments after reading this position papers**

**Rating:** 6
**Confidence:** 2

**Review:**

This position paper pushes for the use of causal models and causal structures to guide the discovery of new concepts in an open world setting. The paper manages to get across the motivation for doing so quite well. I struggled somewhat following the path that the paper outlines towards achieving this goal, but this is challenging to get across clearly in a 2-page paper.

I think the paper connects well with the bridge topics.

---

### Official Review · Reviewer_zHRc · 2022-12-02
**Not entirely clear how to interpret the problem**

**Rating:** 5
**Confidence:** 4

**Review:**

This paper proposes to explore the problem of discovering concepts with background causal knowledge represented in a structural equation model or a causal graph. The idea is that "unknown" concepts may be discovered from known concepts and some causal knowledge that connect the known concepts to "unknown" ones. I am a little puzzled by this formulation of the problem. In what sense is a given concept "unknown" if it is already explicitly featured in the background causal knowledge? Perhaps I misread the formulation in the paper, but that's what I can gather from the current version of the paper.

Judging from the example of match striking in the paper, a task that makes sense to me and might be what the paper tries to describe is that we use background causal knowledge to automatically add new labels. That is, for example, an image that hasn't yet been labelled "flame" can be given this label if matching striking is detected or already a label of the image, based on a given causal rule. I am not sure this is part of what the paper proposes to do. In any case, the paper seems to propose something much more ambitious, which is elusive to me for the reason stated above.

---

### Official Review · Reviewer_2nwU · 2022-12-03
**Good research direction, but maybe difficult to realize**

**Rating:** 6
**Confidence:** 3

**Review:**

It's a good research direction, but it feels difficult to implement.


Pros：
1. I agree that the causal information is helpful in concept discovery. Because  causality and concept are usually closely related in the human cognition.
2. The author gives a possible path to explore the causal information in the concept discovery, including using SCM,  bootstrapping, etc.

Cons:
1. Causal discovery relies on strong assumptions, how to bridge the gap between strong assumptions and the open world? Please provide some discussions.
2. Concept discovery should have a strong connection with causal representation learning. Some related work and research directions should be mentioned and discussed.

---

### Decision · Program_Chairs · 2022-12-05

**Decision:**

Accept

**Comment:**

Accept - Poster

The paper discusses the problem of concept discovery in an open-world setting using the framework of causality. The problem is well-motivated and connects well to the bridge program. However, the reviewers struggled somewhat to follow the ideas and concepts presented in the paper. Moreover, they perceived the proposed solutions as too ambitious and hard to implement. We recommend the authors read the reviews carefully and prepare a camera-ready version of the paper with a clear description of the problem and a discussion of the implications of the underlying assumptions.